# Low Dose of IL-2 Normalizes Hypertension and Mitochondrial Function in the RUPP Rat Model of Placental Ischemia

**DOI:** 10.3390/cells10102797

**Published:** 2021-10-19

**Authors:** Evangeline Deer, Lorena M. Amaral, Nathan Campbell, Sarah Fitzgerald, Owen Herrock, Tarek Ibrahim, Babbette LaMarca

**Affiliations:** 1Department of Pharmacology and Toxicology, University of Mississippi Medical Center, Jackson, MS 39216, USA; edeer@umc.edu (E.D.); lmamaral@umc.edu (L.M.A.); ncampbell@umc.edu (N.C.); sjfitzgerald@umc.edu (S.F.); oherrock@umc.edu (O.H.); ibrahimt@uthscsa.edu (T.I.); 2Department of Obstetrics and Gynecology, University of Mississippi Medical Center, Jackson, MS 39216, USA; 3Departments of Pharmacology, Physiology, and Obstetrics and Gynecology, Center for Excellence in Cardiovascular and Renal Research, University of Mississippi Medical Center, Jackson, MS 39216, USA

**Keywords:** hypertension, preeclampsia, IL-2, inflammation, oxidative stress, placental ischemia

## Abstract

IL-2 is a cytokine released from CD4+T cells with dual actions and can either potentiate the inflammatory response or quell a chronic inflammatory response depending on its circulating concentration. IL-2 is elevated in many chronic inflammatory conditions and is increased during preeclampsia (PE). PE is characterized by new-onset hypertension during pregnancy and organ dysfunction and increasing evidence indicates that proinflammatory cytokines cause hypertension and mitochondrial (mt) dysfunction during pregnancy. The reduced uterine perfusion pressure (RUPP) model of placental ischemia is a rat model of PE that we commonly use in our laboratory and we have previously shown that low doses of recombinant IL-2 can decrease blood pressure in RUPP rats. The objective of this study was to determine the effects of a low dose of recombinant IL-2 on multi-organ mt dysfunction in the RUPP rat model of PE. We tested our hypothesis by infusing recombinant IL-2 (0.05 ng/mL) into RUPP rats on GD14 and examined mean arterial pressure (MAP), renal, placental and endothelial cell mt function compared to control RUPP. MAP was elevated in RUPP rats (*n* = 6) compared to controls (*n =* 5) (122 ± 5 vs. 102 ± 3 mmHg, *p* < 0.05), but was reduced by administration of LD recombinant IL-2 (107 ± 1 vs. 122 ± 5 mmHg, *n =* 9, *p* < 0.05). Renal, placental and endothelial mt ROS were significantly increased in RUPP rats compared to RUPP+ IL-2 and controls. Placental and renal respiration rates were reduced in RUPP rats compared to control rats but were normalized with IL-2 administration to RUPPs. These data indicate that low-dose IL-2 normalized multi-organ mt function and hypertension in response to placental ischemia.

## 1. Introduction

Preeclampsia (PE) is a pregnancy associated disorder affecting 5–7% of pregnancies worldwide and is a well-known cause of maternal, fetal, neonatal morbidity and mortality [1]. PE is defined as new-onset hypertension and end-organ dysfunction during pregnancy occurring after the 20th week of gestation and is associated with chronic immune activation, proteinuria, fetal growth restriction and maternal endothelial dysfunction [2]. The only treatment for PE is delivery of the fetus, which is oftentimes pre-term. Therefore, additional investigation into the pathophysiological mechanisms that lead to the development of PE is necessary in order to develop potential therapies.

A normal pregnancy evolves with tightly controlled immune responses, whereas pregnancies diagnosed as PE exhibit a heightened pro-inflammatory immune response [3]. The complex immune response in PE has been associated with inflammatory immune cells and cytokines, which leads to the production of reactive oxygen species (ROS), increased expression of endothelin-1 (ET-1), sFlt-1 and autoantibodies to the angiotensin II type 1 receptor (AT1-AA) [4,5]. Importantly, in PE, reduced uterine perfusion may cause placental ischemia, a phenomenon that has been well demonstrated in the reduced uterine perfusion pressure (RUPP) rat model of preeclampsia [2,5,6,7]. Reactive oxygen species (ROS) are highly reactive free radicals that damage DNA, RNA, and protein, leading to cellular dysfunction and death. Oxidative stress is created during the imbalance between ROS and antioxidant defense of the cell [6,8]. In normal pregnancies, there is an increase in ROS compared to the non-pregnant state; however, ROS production is excessive in pathological states such as preeclampsia [6,9]. Vaka [6] examined mitochondrial (mt) dysfunction and ROS in the RUPP rat model of PE and found that mt dysfunction contributed to the hypertension. Although we know the importance of renal and placental mt dysfunction in hypertension in the RUPP model of PE, other avenues to lower excessive ROS or improve mt function need to be examined.

Many clinical studies have demonstrated that low-dose IL-2 ranging from 0.3 × 10^6^ to 3.0 × 10^6^ IU improved chronic inflammatory states and outcomes in patients with type 1 diabetes, ischemic heart disease, autoimmune liver disease, and lupus [10,11,12,13]. We have recently shown that low-dose IL-2, specifically (LD = 0.01 IU: 0.05 ng/mL), attenuated circulating and placental NK cells, normalized T regulatory cells, and lowered sFlt-1 and renal preproendothelin and blood pressure in the RUPP rats [14]. However, we do not know the effect of IL-2 on renal or placental mt dysfunction as a mechanism to improve hypertension. Therefore, we repeated our study and infused low-dose IL-2 (0.01 IU) into the RUPP rats and evaluated its effect on blood pressure and multi-organ mt function.

## 2. Materials and Methods

Timed-pregnant 12-week-old female Sprague Dawley (SD) rats (>240 g) were purchased from Envigo (Indianapolis, IN, USA) housed in an enclosed temperature-controlled room (75 °F) consisting of a 12:12 h light/dark cycle and free access to standard chow and water. All experiments were in compliance with the guidelines of the University of Mississippi Medical Center, and the animals were handled with care based on the approved protocol #1435 (12/1/2020) and published principles in the National Institutes of Health Guide for the Care of Animals and the Institutional Animal Care and Use Committee (IACUC).

Rats were divided into three groups consisting of normal pregnant rats (NP, *n =* 5), reduced uterine perfusion pressure rats (RUPP, *n =* 6), and RUPP rats treated with a low-dose (LD) treatment of recombinant IL-2 (RUPP + LD IL-2, *n =* 9, 0.05 ng/mL) (Recombinant IL-2, R and D Systems, Minneapolis, MN, USA). On day 14, the RUPP surgery was performed [15]. This surgical procedure is a model of preeclampsia in the rat and mirrors the pathophysiology of PE in women [6,15]. Surgical clips were placed just above the iliac bifurcation on the abdominal aorta and on ovarian arteries on the left and right side to reduce blood flow by approximately 40%. One group of pregnant RUPP rats received a low dose of recombinant IL-2 (0.05 ng/mL) infused intraperitoneally by a mini-osmotic pump (Alzet; Model 2002; Cupertino, CA, USA) inserted on day 14 of pregnancy. We have previously published that infusion of this dose of recombinant IL-2 into normal pregnant rats had no effect on blood pressure [14]. On day 18, all groups were inserted with indwelling carotid catheters [15]. Following the RUPP procedure, analgesics were provided to the rats, and included 5 mg/kg carprofen administered via subcutaneous injection and once daily for 2–3 days following RUPP surgical procedure, and 0.25% bupivacaine hydrochloride administered topically after carotid catheter insertion. On GD 19, blood pressure was measured with a pressure transducer (Cobe II tranducer CDX Sema, Aurora, CO, USA) and recorded continuously for one hour after a 30 min stabilization period as previously described [15], and is the average of several of the readings over a one-hour period. On gestation day 19, mean arterial pressure (MAP), fetal and placental weights were measured, and blood, placentas, and kidneys were collected for analysis of mitochondrial function.

### 2.1. Isolation of Mitochondria

Renal or placenta mitochondria were isolated from rats using differential centrifugation method [6,16]. Concisely, fresh tissues were rinsed and processed using a dounce homogenizer. The homogenate was centrifuged at 4000 rpm for 3 min at 4 °C. The supernatant was centrifuged at 10,000 rpm for 10 min at 4 °C, and the pellet was collected and suspended in 1 mL of Mito I buffer (250 mM sucrose, 10 mM HEPES, 1 mM EGTA 0.1% BSA, pH 7.2) and centrifuged at 10,000 rpm for 10 min at 4 °C. The collected pellet was suspended in 1 mL of Mito II (250 mM sucrose, 10 mM HEPES, 0.1% BSA, pH 7.2) and centrifuged at 10,000 rpm for 10 min at 4 °C. The final pellet was collected and suspended in 200 µL of Mito II buffer and used for respiration and ROS experiments.

### 2.2. Mitochondrial Respiration

Respiration in isolated mitochondria was measured using an Oxygraph 2K. The basal, state 2, state 3, state 4, and uncoupled respiration rates were measured using glutamate/malate, ADP, oligomycin, and FCCP (carbonyl cyanide-4-[trifluoromethoxy]phenylhydrazone), respectively [6]. Non-mitochondrial respiration was recorded with the use of Rotenone and antimycin A. The data collected were analyzed and expressed as pmol of oxygen consumed per second per milligram of mitochondrial protein.

### 2.3. Mitochondrial ROS

Mitochondrial hydrogen peroxide (H_2_O_2_) production in placental and renal mitochondria was determined by using amplex red assay [6,17]. Mitochondria (0.4 mg/mL) were incubated in a 96 well plate containing respiration buffer, superoxide dismutase (40 U/mL), horseradish peroxidase (4 U/mL), and succinate (10 mM). Amplex red (10 µM) was added to the wells last to start the reaction. The final volume of the wells used in the microplate was 200 µL. The real-time production of H_2_O_2_ was measured at 555/581 nm excitation/emission using a plate reader for 30 min at 25 °C. Sample controls (blanks without mitochondrial protein or amplex red) were included in the assay.

### 2.4. Endothelial Mitochondrial ROS

Mitochondrial-specific reactive oxygen species were measured using MitoSOX red, a fluorogenic dye that specifically targets the mitochondria in live cells. HUVECs (ATCC), passage 4, were grown to 70% confluency in 6 well culture plates in HUVEC complete growth medium [Medium 199-DMEM (50:50), 10% FBS, and 1% antimycotic/antibiotic] in a humidified atmosphere of 5% CO_2_ at 37 °C. Cells were serum starved for 4 h prior to incubation with HUVEC complete growth media and 10% of individual serum from NP (*n =* 5), RUPP (*n =* 4), or RUPP + LD IL-2 (*n =* 7) sera overnight. Each experiment for individual rats were performed in duplicate and averaged together/animal. The data were then averaged for each group.

Media with serum was rinsed off and cells were incubated with MitoSOX red (5 µM) for 30 min at 37 °C. Antimycin A (100 µM) was utilized as a positive control. Serum free medium was added after washing the cells twice with DPBS and the cells were incubated for an additional 4 h. Cells were collected and analyzed in the FL2 channel of Gallios flow cytometer (Beckman Coulter, Brea, CA, USA).

### 2.5. IL-2 Cytokine Profile

Blood samples from NP, RUPP, and RUPP + LD IL-2 were collected and centrifuged at 825× *g* for 10 min at 4 °C. Serum samples were separated from the clot. The supernatant was obtain, aliquoted, and stored at −80 °C. IL-2 cytokine levels in serum samples from NP, RUPP, and RUPP + LD IL-2 were measured using a Bio-Plex immunoassay according to the manufacturer’s instructions. Data were acquired using the BIO-PlexTM 200 system (Bio-Rad Laboratories, Hercules, CA, USA).

### 2.6. Statistical Analysis

All statistical analyses were performed with GraphPad Prism 7.02 software (GraphPad Software, San Diego, CA, USA). Results were reported as means ± SEM. Comparison of groups were assessed by one-way ANOVA with Bonferroni multiple comparisons test as post hoc analysis. Results were considered as statistically significant when *p* < 0.05.

## 3. Results

### 3.1. IL-2 Significantly Lowered Blood Pressure in RUPP Rats

Mean arterial pressure (MAP) was elevated in RUPP rats (*n =* 6) compared to NP controls (*n =* 5) (122 ± 5 vs. 102 ± 3 mmHg, *p* < 0.05), but was reduced by administration of LD IL-2 in RUPP rats (107 ± 1 vs. 122 ± 5 mmHg, *n =* 9, *p* < 0.05) (Figure 1). Placental weights were reduced in both RUPP rats (0.53 ± 0.03 g, *p* < 0.05) and RUPP + LD IL-2 rats (0.50 ± 0.02 g, *p* < 0.05) compared to NP controls (0.66 ± 0.04 g) (Table 1). Fetal weights were reduced in RUPP rats (1.99 ± 0.07 g, *p* < 0.05) and RUPP + IL-2 rats (1.95 ± 0.08 g, *p* < 0.05) compared to NP controls (2.27 ± 0.05 g) (Table 1). Percent reabsorptions and survivability were reduced in RUPP rats compared to NP controls and RUPP + LD IL-2 rats (Table 1).

### 3.2. IL-2 Significantly Improved Multi-Organ mt Function in RUPP Rats

Placental mitochondrial ROS, as measured by production of H_2_O_2_, was significantly elevated in RUPP rats (144.6 ± 14.18% gated, *p* < 0.05, *n =* 5) compared to NP controls (100 ± 12.34% gated, *n =* 5), but was normalized in RUPP + LD IL-2 rats (108.7 ± 7.38% gated, *p* < 0.05, *n =* 9) (Figure 2A). Renal mitochondrial ROS increased in real-time production of H_2_O_2_ in RUPP rats (127.1 ± 2.81% gated, *p* < 0.05, *n =* 5) in comparison to NP controls (100 ± 5% gated, *n =* 5), but was significantly reduced in RUPP + LD IL-2 rats (63.26 ± 3.57% gated, *p* < 0.05, *n =* 9), Figure 2B.

Placental mitochondrial state 3 respiration, which is indicative of ATP produced from the addition of ADP, significantly decreased in RUPP rats (*n =* 6) (24.83 ± 15.53 pmol of O_2_/s/mg, *p* < 0.05) compared to NP controls (*n =* 5) (132.9 ± 6.64 pmol of O_2_/s/mg) (Figure 3A). State 3 placental respiration increased significantly in RUPP rats + LD IL-2 (*n =* 6) (157.3 ± 48.56 pmol of O_2_/s/mg, *p* < 0.05) compared to RUPP rats, Figure 3A. Placental mitochondrial uncoupled respiration, which is indicative of electron transport chain function, was reduced significantly in RUPP rats (*n =* 6) (14.96 ± 3.89 pmol of O_2_/s/mg, *p* < 0.05) compared to NP controls (*n =* 5) (91.02 ± 15.73 pmol of O_2_/s/mg) (Figure 3B) but was significantly improved in RUPP + LD IL-2 rats (*n =* 6) (118.1 ± 35.42 pmol of O_2_/s/mg, *p* < 0.05) compared to RUPP rats, Figure 3B.

Renal mitochondrial state 3 respiration, which is indicative of ATP produced from the addition of ADP, was significantly reduced in RUPP rats (*n =* 5) (138.4 ± 48.21 pmol of O_2_/s/mg, *p* < 0.05) compared to NP controls (*n =* 5) (958 ± 200.6 pmol of O_2_/s/mg) (Figure 4A), but was normalized in RUPP + LD IL-2 rats (*n =* 6) (904 ± 288 pmol of O_2_/s/mg, *p* < 0.05), Figure 4A. Renal mitochondrial uncoupled respiration, which is indicative of electron transport chain function, decreased significantly in RUPP rats (*n =* 5) (68.1 ± 9.29 pmol of O_2_/s/mg, *p* < 0.05) compared to NP controls (*n =* 5) (476 ± 95.3 pmol of O_2_/s/mg) (Figure 4B), but was increased in RUPP + LD IL-2 rats (*n =* 6) (824 ± 255.3 pmol of O_2_/s/mg, *p* < 0.05) compared to RUPP, Figure 4B.

Placental RCR (state 3/state 4) significantly decreased in RUPP rats (*n =* 6) (0.79 ± 0.21 RCR, *p* < 0.05) compared to NP controls (*n =* 5) (1.77 ± 0.25 RCR) (Figure 5A), but was normalized in RUPP + LD IL-2 rats (*n =* 6) (1.77 ± 0.25 RCR, *p* < 0.05), Figure 5A. Although there was a decrease in renal RCR in RUPP rats (1.23 ± 0.37 RCR, *n =* 5) compared to NP controls (2.07 ± 0.35 RCR, *n =* 5), it was not significant nor was renal RCR improved in RUPP + LD IL-2 rats (1.30 ± 0.13 RCR, *n =* 5), Figure 5B.

Mt ROS significantly increased in endothelial cells, HUVECS, exposed to media containing sera from RUPP (*n =* 4) (6.38 ± 1.81% gated, *p* < 0.05) compared to NP controls (*n =* 5) (1.86 ± 0.6% gated) (Figure 6), but was normalized in RUPP + IL-2 rats (*n =* 7) (2.69 ± 0.53% gated, *p* < 0.05) compared to RUPP, Figure 6.

### 3.3. Serum IL-2 Levels Are Increased in PE

As shown in Figure 7, IL-2 levels were higher RUPP + LD IL-2 (68 ± 16 pg/mL, *p* < 0.05) than RUPP control rats (5 ± 3.5 pg/mL, *n =* 4).

## 4. Discussion

One hallmark of PE is multi-organ dysfunction, which can include a combination of renal, hepatic, neural, cardiac, placental or endothelial dysfunction. Appropriate cellular processes at the level of the mitochondria such as maintaining electron transfer, cellular respiration, and oxygen utilization are important for tissue homeostasis and function. In this study, we investigated the effects of recombinant IL-2 supplementation on PE characteristics such as hypertension and placental and renal mt dysfunction/ROS in RUPP rats. In addition, we measured endothelial mt dysfunction from cells exposed to circulating factors in sera from control normal pregnant rats, RUPP rats and RUPP rats + IL-2. The RUPP rat model of placental ischemia is a well characterized and well known model of preeclampsia that mimics the physiological features of humans, including hypertension, immune system abnormalities, systemic and renal vasoconstriction, and oxidative stress in the mother, and intrauterine growth restriction found in the offspring. Therefore, the RUPP model of placental ischemia has been shown by our lab and others as a useful model in studying the effects of placental ischemia. Although a limitation of our study is that we had a small number of animals in our experimental groups, the overall purpose of our study was to understand more about the effect of IL-2 and its role in inflammation in response to placental ischemia. The results of our study demonstrated that IL-2 normalized mean arterial pressure and significantly reduced the mt ROS in the kidney and placenta of RUPP rats. Moreover, IL-2 reduced mt ROS from HUVECs exposed to sera from RUPP rats, indicating that IL-2 was able suppress circulating factors produced in response to placental ischemia that stimulate endothelial dysfunction. Importantly, renal and placental respiration were reduced in RUPP rats compared to normal pregnant rats. Low-dose recombinant IL-2 was able to normalize both renal and placental respiration, thus indicating that low-dose IL-2 was able to improve organ function in response to placental ischemia.

A compilation of research indicates that endothelial and mt dysfunction is readily observed in preeclamptic placentas [6,18,19,20,21,22,23,24,25,26,27]. Previously, McCarthy [28] showed vascular mtROS and decreased respiration in HUVECs exposed to sera from PE patients compared to HUVECS cultured with sera from normal pregnant women, thus indicating the importance of the release of soluble factors in the circulation to cause cellular mt dysfunction. Notably, we demonstrated that the blockade of circulating AT1-AA from human PE sera was able to attenuate mtROS in HUVECS cultured with PE sera with and without AT1-AA blockade [23]. In addition to the AT1-AA, our lab has recently investigated the importance of mt oxidative stress in PE pathology, and has linked reduced vascular endothelial mt respiration and mtROS with the presence of CD4+ T cells stimulated in response to placental ischemia [23,24,25,26,27,28,29]. In a previous study from our groups, we showed that LD recombinant IL-2 improved T regs cell in RUPP rats. In corroboration with our previous studies, this study demonstrated that infusion of low-dose IL-2 was able to normalize impaired mitochondrial function in tissues and vascular endothelium, thereby presenting a novel role for IL-2, possibly via improving T reg cells, in the pathology of preeclampsia. Importantly, PE is caused by multiple factors, and we investigate the role of various factors to contribute to the pathology of the disease and to possibly eliminate gaps in the literature. Although this study investigated IL-2, there is still more to understand about preeclampsia and the other factors that contribute to this disease. In addition, multiple factors can induce placental mitochondrial ROS production, and for example, we have previously shown that natural killer cells [4] and CD4+ T cells [29] cause mitochondrial dysfunction in RUPP rats and endothelial cells incubated with RUPP serum exhibit mitochondrial ROS [6]. HUVECS treated with serum from PE women and incubated with MitoSOX Red indicated that serum from PE women contained circulating factors which contribute to mitochondrial dysfunction and an increase in mt ROS in cultured human vascular endothelial cells, thereby demonstrating that an increase in oxidative stress contributes to endothelial dysfunction [6,23]. Furthermore, PE is associated with chronic immune activation, thereby leading to an increase in inflammatory cytokine production. This imbalance leads to chronic inflammation that is characterized by increases in pro-inflammatory cytokines and oxidative stress, (ROS), (endothelin (ET-1), and agonistic autoantibodies to the angiotensin II (Ang II), type 1 receptor (AT1-AA). These and other factors may influence mitochondrial activity and its contribution to maintenance of normal blood pressure during pregnancy. Discovering strategies that could potentially target mitochondrial stress via reducing mt oxidative stress and improving mitochondrial function is a finding that will greatly benefit maternal and fetal outcomes.

We recently showed a low-dose regimen of IL-2 consisting of three regimens between 0.01 and 0.05 IU into the RUPP rats significantly increased T Regs and decreased NK cells and hypertension during pregnancy [14]. We chose the lowest dose IL-2 from the previous study we performed because it had least detrimental effects on pup weight and survivability while still lowering the blood pressure and other factors associated with placental ischemia of pregnancy, such as sFlt-1 and renal endothelin-1. Moreover, this dose of IL-2 normalized circulating T regs in our previous study. Considering the importance of IL-2 in NK cell and T cells maturation, transformation and activity, we utilized this dose for our current study.

Regulatory T cells are necessary to maintain an immune steady state and to prevent autoimmune diseases. IL-2 has been touted for its unique ability for T cell expansion, function, and survival. When disrupted, IL-2-dependent balance of Treg and T effector cells causes autoimmunity and chronic inflammation. Recent studies have indicated that treatment with a low-dose IL-2 induces immune tolerance resulting in the suppression of an unwarranted immune responses and suggests that it may be a possible treatment of certain autoimmune disorders [30]. Because Treg cells cannot make their own IL-2, they depend on IL-2 produced by activated CD4+ T cells, therefore, linking T effector and T reg cell populations for immune homeostasis. In numerous autoimmune diseases, there is a decrease in the numbers and function of Treg cells [31,32,33,34] that is restored by exogenous low doses of IL-2 in mice [35,36] and humans [37,38,39]. Clinical trials of low-dose IL-2 in patients have had selective effects on T regs in healthy individuals, patients with hepatitis C virus-induced vasculitis, type 1 diabetes, and systemic lupus erythematosus [39,40,41,42]. Although some studies have reported that IL-2 did not alter or increase blood pressure in rats, it appears that the differences in the dose and frequency of IL-2 injection may be responsible for the difference in the lack of antihypertensive effect.

Previous studies have shown that IL-2 attenuated progression of hypertension in Dahl S rats, which was accompanied by improvements in renal dysfunction and cardiac hypertrophy [43,44]. In addition, treatment with IL-2 immune complex of hypertensive mice was shown to increase T regs and reduce aortic stiffening [44]. Moreover, a low dose of IL-2 administered in mice was able to prevent type 1 diabetes mellitus and improve the numbers of Tregs via their programming dependence on IL-2 [45]. Importantly, a high dose of IL-2 could produce lethal toxicity [46,47] and lead to the destruction of cells making high-dose IL-2 efficacious in treating metastatic cancer due to increasing the activity of natural killer cells toward the tumors. Therefore, the dose of IL-2 appears to be a determining factor in the imbalance between immune tolerance and destructive autoimmunity and is important to be established safely before moving forward as a treatment for any disease.

In order to successful maintain pregnancy, cytokines are necessary for maintaining a fetotolerant environment. Although IL-2, TNF-α, and IFN-γ are characteristic of T Helper 1-type immunity and induce a cytotoxic and an inflammatory reaction, IL-2 has been shown to improve inflammation during pregnancy without causing any detrimental effects [48]. In a normal pregnancy, IL-2 levels are decreased to concentrations necessary for the development, proliferation, and survival of T regs [49,50]. When IL-2 levels are high, pregnant women have a higher susceptibility for spontaneous abortion, preterm delivery, IUGR, and the development of PE [51,52]. High doses of IL-2 cause capillary damage, renal and liver damage, and hypotension [10,11,12]. Yet, higher doses of IL-2 have been coupled with additional drugs as a chemotherapeutic agent [10,11,12,53]. At high doses, NK cell proliferation and cytolytic activity are stimulated, which is an important component of some metastatic cancer therapies [53]. Furthermore, a high-dose treatment of IL-2 has been shown to be beneficial in treating metastatic cancer because of an increased activity of natural killer cells towards tumors. Therefore, high-dose treatments have been more common and utilized for longer periods of treatment compared to those utilizing lower doses of IL-2. However, these current studies show that in late pregnancy, low-dose IL-2 may help to lessen inflammation in response to placental ischemia which in turn lowers hypertensive molecules such as sFlt-1, ET-1 and mt ROS.

Cytolytic natural killer cells were also significantly decreased with LD IL-2 infusion in to RUPP and normal pregnant rats [14]. Because the immune cell profile was so very different between RUPP and RUPP+IL-2 in our previous study and because we have shown the importance of both T cells and NK cells to cause mitochondrial dysfunction, we examined the effects of IL-2 on mt function in RUPP rats. Mitochondria are an important source of ROS production, and the superoxide that is not able escape the mitochondria is reduced to hydrogen peroxide. Therefore, the highly reactive free radicals (ROS) damage the cellular contents and result in cellular dysfunction and cell death. Increased cell death within a tissue contributes to an overall dysfunction of that tissue and organ and therefore damage to mitochondrial function correlates with dysfunctional organ systems. HUVECS supplemented with serum from preeclamptic and normotensive women have shown impaired tube like structure formation and normal regular tube-like structure, respectively [9]. Furthermore, an in vitro model of HUVECS treated with PE serum demonstrates that NADPH oxidase activity is increased and thus is important in O_2_ formation [54,55,56]. We have shown that circulating factors such as the AT1-AA and TNF-α contribute to vascular endothelial cell mt dysfunction. In this study, we show that LD IL-2 improves not only renal and placental mt function in RUPP rats but that circulating factors stimulated by placental ischemia in the RUPP treated with LD IL-2 are decreased to the extent that RUPP sera no longer stimulates endothelial cell mt dysfunction and ROS.

## 5. Conclusions

In conclusion, supplementation of IL-2 significantly decreased the blood pressure in RUPP rats and lowered both placental and renal mt dysfunction/ROS and endothelial mt dysfunction/ROS. Moreover, in the current study, infusing a low dose of IL-2 mitigated the decrease in fetal reabsorptions, and thereby increased the percent survivability of the fetus in contrast to untreated RUPP rats. Although the previous study by Cunningham et al. [14] showed adverse fetal effects of LD IL-2, this study did not, which could be due to different operator’s skills, yet both studies did demonstrate an increase in reabsorptions compared to NP controls. Therefore, coupled with our previous study, these results demonstrate that IL-2 normalized mt function in RUPP rats, which is associated with lower blood pressure and improved fetal survivability, thereby indicating that a potential therapeutic target for PE could be a carefully planned regimen of LD IL-2.

## Figures and Tables

**Figure 1 cells-10-02797-f001:**
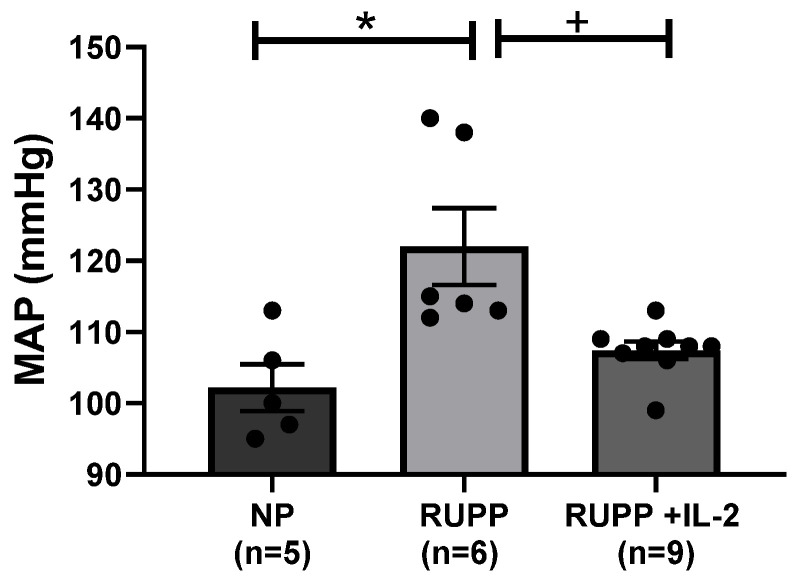
Mean arterial pressure was elevated in RUPP rats (*n =* 6) compared to NP rats (*n =* 5), but was normalized in RUPP rats administered a lose dose of IL-2 (*n =* 9). Results were reported as means ± SEM and considered statistically significant when *p* < 0.05. (* *p* < 0.05 vs. NP control; + *p* < 0.05 vs. RUPP).

**Figure 2 cells-10-02797-f002:**
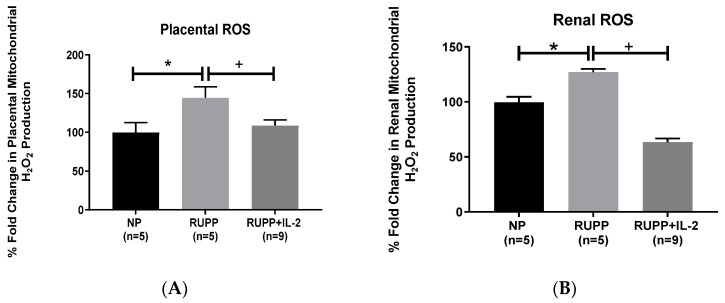
There were increases in the (**A**) placental and (**B**) renal production of mitochondrial ROS in RUPP rats (*n =* 5) compared to NP rats (*n =* 5). Administration of a low dose IL-2 in RUPP rats (*n =* 9) normalized the production of mitochondrial ROS in the placenta and kidney. Results were reported as means ± SEM and considered statistically significant when *p* < 0.05. (* *p* < 0.05 vs. NP control; + *p* < 0.05 vs. RUPP).

**Figure 3 cells-10-02797-f003:**
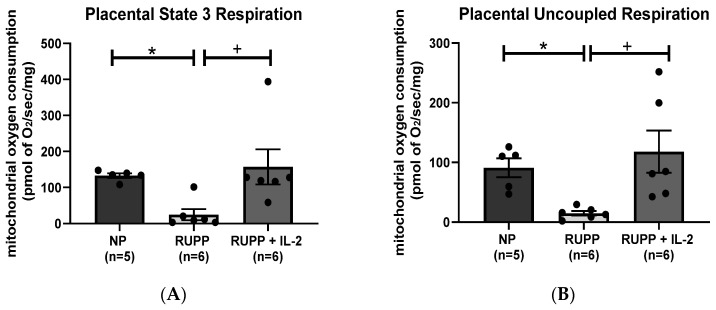
There reductions in (**A**) state 3 and (**B**) uncoupled placental mitochondrial respiration in RUPP rats (*n =* 6) compared to RUPP + LD IL-2 (*n =* 6) and NP rats (*n =* 5). Results were reported as means ± SEM and considered statistically significant when *p* < 0.05. (* *p* < 0.05 vs. NP control; + *p* < 0.05 vs. RUPP).

**Figure 4 cells-10-02797-f004:**
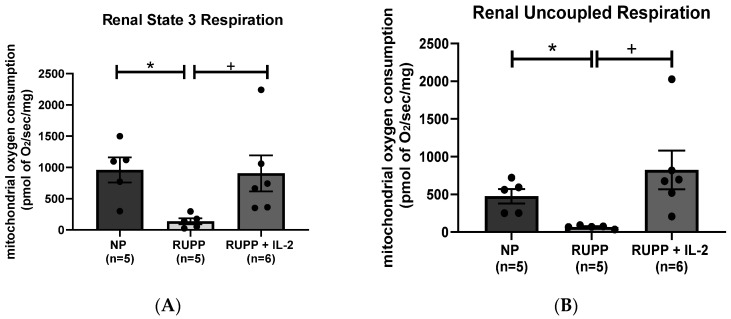
(**A**) State 3 and (**B**) uncoupled renal mitochondrial respiration were reduced in RUPP rats (*n =* 5) compared to NP rats (*n =* 5), but were normalized in RUPP + LD IL-2 rats (*n =* 6). Results were reported as means ± SEM and considered statistically significant when *p* < 0.05. (* *p* < 0.05 vs. NP control; + *p* < 0.05 vs. RUPP).

**Figure 5 cells-10-02797-f005:**
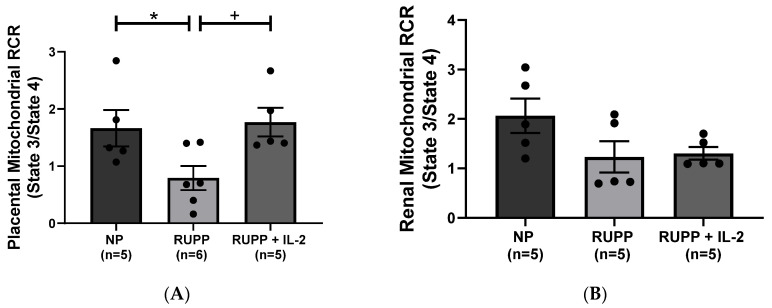
(**A**). Placental RCR (state 3/state 4) was reduced in RUPP (*n =* 6) compared to both NP rats (*n =* 5) and RUPP + LD IL-2 rats (*n =* 6). (**B**). Renal RCR (state3/state 4) was reduced in RUPP rats (*n =* 5) compared to NP rats (*n =* 5).There was no significant difference in RCR demonstrated in RUPP + LD IL-2 rats (*n =* 5) compared to RUPP rats. Results were reported as means ± SEM and considered statistically significant when *p* < 0.05. (* *p* < 0.05 vs. NP control; + *p* < 0.05 vs. RUPP).

**Figure 6 cells-10-02797-f006:**
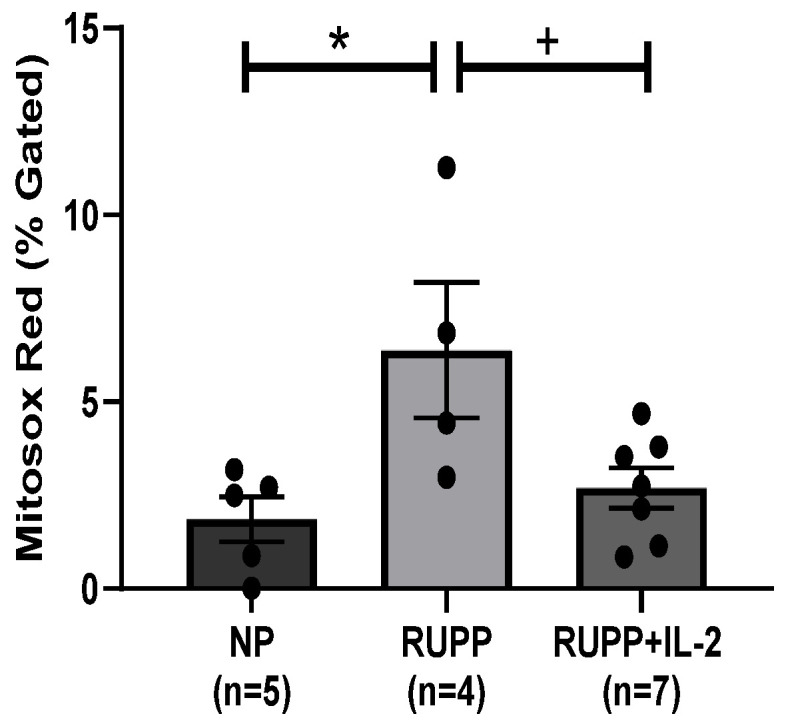
HUVECS incubated with RUPP sera (*n =* 4) exhibited an increase in endothelial mt dysfunction as demonstrated by an increase in mtROS compared to NP sera (*n =* 5), but co-incubation with RUPP + LD IL-2 sera (*n =* 7) normalized the production of mtROS. Results were reported as means ± SEM and considered statistically significant when *p* < 0.05. (* *p*< 0.05 vs. NP control; + *p* < 0.05 vs. RUPP).

**Figure 7 cells-10-02797-f007:**
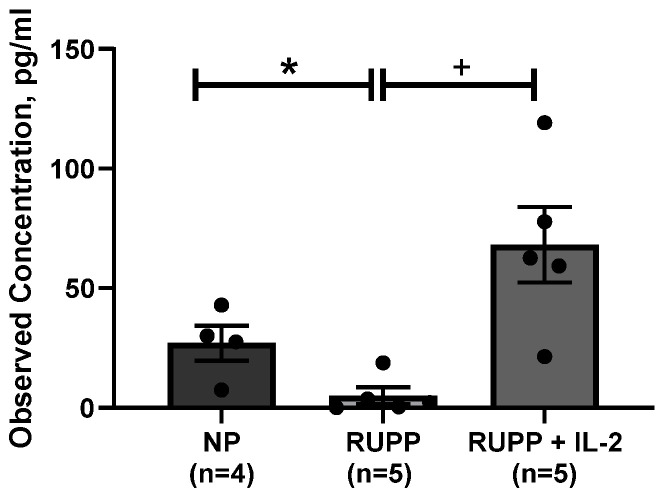
Comparison of IL-2 cytokine levels demonstrated that RUPP sera (*n =* 5) was lower compared to RUPP rats treated with a low dose of recombinant IL-2 (*n =* 5) and NP controls sera (*n =* 4). Results were reported as means ± SEM and considered statistically significant when *p* < 0.05. (* *p* < 0.05 vs. NP control; + *p* < 0.05 vs. RUPP).

**Table 1 cells-10-02797-t001:** Placental Weights, Fetal Weights, Percent of Total Reabsorptions, and Percent Survival.

Animal Group	Placental Weight (g)	Fetal Weight (g)	% Reabsorptions	% Survived
NP	0.66 ± 0.04	2.27 ± 0.05	0 ± 0	100 ± 0
RUPP	0.53 ± 0.03 *	1.99 ± 0.07 *	19.6 ± 4 *	80.4 ± 4 *
RUPP + LD IL-2	0.50 ± 0.02 ^+^	1.95 ± 0.08 ^+^	0.43 ± 0.1 ^#^	99.5 ± 0.1 ^#^

* *p* < 0.05 RUPP vs. NP control; ^+^ *p* < 0.05 RUPP + LD IL-2 vs. NP control; ^#^ *p* < 0.05 RUPP + LD IL-2 vs. RUPP.

## Data Availability

All data relevant to the study are included in the article.

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
