# Peer review of "Low Dose of IL-2 Normalizes Hypertension and Mitochondrial Function in the RUPP Rat Model of Placental Ischemia"

_cells, 2021, doi:10.3390/cells10102797_

Round 1

Reviewer 1 Report

In this study, the authors investigated the role of IL-2 treatment in response to blood pressure in a placental ischemia model. It is a very interesting study. However, I have some concerns: 

  1. the sample size of RUPP, I think n=5 or 7 was quite small.
  2. Do the authors have systolic blood pressure instead of MAP? As the definition of preeclampsia is systolic over 140 mmHg. From Figure 1, it seems that the MAP is around 120 mmHg. I am not sure whether this MAP is high enough for defining preeclampsia. Studies suggested that MAP over 125 mmHg is considered in preeclampsia.  Only two RUPP rats showed higher MAP (over 125 mmHg). 
  3. can the authors provide the age of Rats? 
  4. it is not clear when the IL-2 was given. and the dose of IL-2, (0.05ng/ml or 0.05ng/kg)?
  5. it is not clear when the MAP was measured. how many time points are for the MAP?
  6. In figure 6, the authors mentioned that serum from RUPP induced an endothelial cell mitochondrial dysfunction. Did the authors measure the endothelial cell dysfunction? it will be of interest to know serum from the RUPP model has any effect on endothelial cell dysfunction. 
  7. It will be also interesting to know whether other neutralizing antibodies such as IL-6 have a similar function as the study has showed increased levels of IL-6 in preeclampsia and the dysfunction of endothelial cells was blocked by neutralizing IL-6 antibody (DOI: 10.1002/path.2425). 
  8. Due to preeclampsia is not A disease, multiple factors cause this disease. So the limation of this study was that the authors only investigated the inflammatory cytokines, which can not present a whole of the preeclampsia. In addition, a number of factors can induce placental mitochondrial ROS production such as aPL (Zussman R 2020). So the authors must address this point.  
  9. In the abstract, cam the authors mention the IL-2 is recombinant IL-2 

Author Response

Reviewer 1 Comments:

  1. the sample size of RUPP, I think n=5 or 7 was quite small.

Response: We appreciate the comment but results with an n=5 was plenty for statistically significance, therefore we did not see a reason to add more to the groups and continue the use of pregnant rats.

  1. Do the authors have systolic blood pressure instead of MAP? As the definition of preeclampsia is systolic over 140 mmHg. From Figure 1, it seems that the MAP is around 120 mmHg. I am not sure whether this MAP is high enough for defining preeclampsia. Studies suggested that MAP over 125 mmHg is considered in preeclampsia.  Only two RUPP rats showed higher MAP (over 125 mmHg). 

Response:We appreciate the comments of the reviewer. The RUPP model has been previously defined and used by many other investigators as a model of PE and consistently exhibits 20-30 mmHg above a normal pregnant rat. As such our focus for this study was not to demonstrate the RUPP as a model of PE as that has been done by others but instead to further examine the role of IL-2 to lower blood pressure and restore a more normal mitochondrial function.

  1. can the authors provide the age of Rats? 

Response: Thank you for your response. Timed-pregnant 12 week old female Sprague Dawley (SD) rats (>240g) were purchased from Envigo (Indianapolis, IN)

  1. it is not clear when the IL-2 was given. and the dose of IL-2, (0.05ng/ml or 0.05ng/kg)?

Response: Thank you for your response. One group of pregnant RUPP rats received a low dose of recombinant IL-2 (0.05 ng/ml) infused intraperitoneal by a mini-osmotic pump (Alzet; Model 2002) inserted on day 14 of pregnancy. This is more clearly stated in the manuscript.

  1. it is not clear when the MAP was measured. how many time points are for the MAP?

Response: Thank you for your response. On gestation day 19, mean arterial pressure (MAP), fetal and placental weights were measured. This is more clearly stated in the manuscript.

  1. In figure 6, the authors mentioned that serum from RUPP induced an endothelial cell mitochondrial dysfunction. Did the authors measure the endothelial cell dysfunction? it will be of interest to know serum from the RUPP model has any effect on endothelial cell dysfunction. 

Response: Thank you for the response. We have previously published this technique showing that RUPP sera causes endothelial cell dysfunction with an increase in ET-1.

Lyndsay R et.al., Hypertension. 2006 Mar;47(3):615-8.

doi: 10.1161/01.HYP.0000197950.42301.dd. Epub 2006 Jan 3.

Enhanced endothelin synthesis by endothelial cells exposed to sera from pregnant rats with decreased uterine perfusion

  1. It will be also interesting to know whether other neutralizing antibodies such as IL-6 have a similar function as the study has showed increased levels of IL-6 in preeclampsia and the dysfunction of endothelial cells was blocked by neutralizing IL-6 antibody (DOI: 10.1002/path.2425). 

Response: Thank you. Although we did not study IL-6 in this particular study, this will be something that we definitely would look at in the future.  

  1. Due to preeclampsia is not A disease, multiple factors cause this disease. So the limitation of this study was that the authors only investigated the inflammatory cytokines, which can not present a whole of the preeclampsia. In addition, a number of factors can induce placental mitochondrial ROS production such as aPL (Zussman R 2020). So the authors must address this point.  

Response: Thank you for this point we have addressed this point in the discussion.

  1. In the abstract, can the authors mention the IL-2 is recombinant IL-2 

Response: Thank you. The abstract has been changed to mention that IL-2 is recombinant IL-2.

Reviewer 2 Report

This study follows on from the previously published study in BOSD (Ref 14.) however it builds on the story by looking at the involvement of mitochondrial function in IL-2 (low does) mechanism of action in the RUPP model of preeclampsia. The study is interesting however I still cannot see this being translated to clinical practica given the adverse effects demonstrated on the placental and pupp weight despite improvement in fetal reabsorption that was previously observed. Also there are aspects of the manuscript writing that can be improved.

Therefore I have a few concerns:

Introduction: it was stated that antioxidants are not recommended during pregnancy? DAPIT (diabetes and preeclampsia interventional study) used vitamin C and E (safe supplements in pregnancy) to investigate the effect on preeclampsia, which was significant in those women with low antioxidant level. Therefore the authors should reframe this sentence as there are many safe antioxidant treatments that can be used in pregnancy whereas IL-2 seems to have adverse effects on the fetus.

Methodology

Were tissue frozen or fresh when processed to isolate mitochondria?

Statistical analysis paragraph states that t-test was used where two groups were compared? I dont see two groups anywhere so the authors should delete this and leave one-way ANOVA only wiht post-hoc tests.

Results

It seems that IL-2 lead to reduction in placental and embryo weight compared to NP but this was not shown in the table? p values comparing 

Why is there less reabsorption in this study compared to Ref 14 published in BOSD with the same IL-2 does? This should be discussed.

Was IL-2 level measured between the NP, RUPP and RUPP+IL-2 animals?

Discussion

Better focus should be on the mitochondrial dysfunction in relation to IL-2 treatment and immune cells. These two aspects should be brought closer together rather than discussed separetly. 

High IL-2 was discussed in the context of causing spontaneous abortion, preterm delivery and other adverse effects. So how does infusion of IL-2 then affect IL-2 levels? 

Why are adverse effects observed on fetus in terms of IUGR? Would even smaller dose than 0.05ng/ml be plausable to have beneficial maternal effects without causing adverse fetal effects?

Future direction regarding translation and clinical relevance of this treatment needs to be discussed. It seems impossible to envisage translation to clinical practice with this treatment at the moment given the adverse effects on the fetus. 

Author Response

Reviewer 2 Comments:

This study follows on from the previously published study in BOSD (Ref 14.) however it builds on the story by looking at the involvement of mitochondrial function in IL-2 (low dose) mechanism of action in the RUPP model of preeclampsia. The study is interesting however I still cannot see this being translated to clinical practice given the adverse effects demonstrated on the placental and pupp weight despite improvement in fetal reabsorption that was previously observed. Also there are aspects of the manuscript writing that can be improved.

Therefore I have a few concerns:

  • Introduction: it was stated that antioxidants are not recommended during pregnancy? DAPIT (diabetes and preeclampsia interventional study) used vitamin C and E (safe supplements in pregnancy) to investigate the effect on preeclampsia, which was significant in those women with low antioxidant level. Therefore the authors should reframe this sentence as there are many safe antioxidant treatments that can be used in pregnancy whereas IL-2 seems to have adverse effects on the fetus.

Response: Revised sentence: Although we know the importance of renal and placental mt dysfunction in hypertension in response to placental ischemia in the RUPP model of PE, other avenues to lower excessive ROS or improve mt function need to be examined. 

  • Methodology: Were tissue frozen or fresh when processed to isolate mitochondria? Statistical analysis paragraph states that t-test was used where two groups were compared? I dont see two groups anywhere so the authors should delete this and leave one-way ANOVA only wiht post-hoc tests.

Response: We have revised sentence to reflect that issues were fresh when processed to isolate mitochondria.

Response: Freshly collected tissues were rinsed, processed using a dounce homogenizer.

Response: We have deleted the t-test and left one-way ANOVA post-hoc tests for statistical analysis.

  • Results: It seems that IL-2 lead to reduction in placental and embryo weight compared to NP but this was not shown in the table? p values comparing . Why is there less reabsorption in this study compared to Ref 14 published in BOSD with the same IL-2 does? This should be discussed. Was IL-2 level measured between the NP, RUPP and RUPP+IL-2 animals?

Response: P-values comparing reduction in placental and embryo weight were added to accurately reflect the results.

Response: IL-2 was measured by Bioplex and this new data was added to demonstrate the level of IL-2 present between the RUPP and RUPP+IL-2 animals.

Response: The reviewer is correct in that the fetal reabsorptions was not similar to Cunningham et al [14]. Overall there is a 20% difference between the reabsorptions/survival rat in response to the RUPP procedure when comparing results between the two studies. We believe this attributed to the different operators of RUPP surgery. This may be the reason for the improved survival rate with IL-2 in the current study as well.

Discussion: Better focus should be on the mitochondrial dysfunction in relation to IL-2 treatment and immune cells. These two aspects should be brought closer together rather than discussed separetly. High IL-2 was discussed in the context of causing spontaneous abortion, preterm delivery and other adverse effects. So how does infusion of IL-2 then affect IL-2 levels? Why are adverse effects observed on fetus in terms of IUGR? Would even smaller dose than 0.05ng/ml be plausable to have beneficial maternal effects without causing adverse fetal effects? Future direction regarding translation and clinical relevance of this treatment needs to be discussed. It seems impossible to envisage translation to clinical practice with this treatment at the moment given the adverse effects on the fetus.

Response: Thank you for your comments comments. We recently showed a Low Dose regimen of IL-2 consisting of 3 regimens between 0.01-0.05 IU into the RUPP rats significantly increased T Regs and decreased NK cells and hypertension during pregnancy [14].   

Reviewer 3 Report

To authors,

The theme is important. Clinical usefulness is yet to be determined, this can be a candidate of therapeutic strategy of PE as the authors suggested. I have some advices.

  1. You stated, “Importantly, these systems occur in response to placental ischemia in the reduced uterine perfusion pressure (RUPP) rat model of preeclampsia. [5 6]. [2 6 7].”. In Introduction, you previously stated/touched “PE general” (mainly in human) and then here, you abruptly change the topic to “rat”. The context is inconsistent. For example, the following may be better (if your intention is so): “In PE, reduced uterine perfusion may cause placental ischemia, of which phenomenon has been well demonstrated in rat model (RUPP rat)”. The same is true to Abstract. Please make the context clear.
  2. Vaka et al. [6]; name and then citation immediately after! There are the similar mistakes here and there.
  3. There have been MANY reported rat models of PE. Please state the reason why you chose this model among many others. Preferably, “to study ROS-related mechanism of PE” “the present model is theoretically better/best”; such context is better as a scientific paper. If not, “we are very accustomed to use this model and thus we used this model” (meaning) may be OK (straightforward statement is welcomed). Anyhow, please describe the reason why you used this model.
  4. You, all through the manuscript, emphasized “placental ischemia”. Please confirm this emphasis is theoretically correct. You used this model; actually, this model causes “placental ischemia”; however, whether this ischemia is a “single” mechanism/reason for the manifestation of PE is not demonstrated. The title also emphasizes “ischemia”. To show “a direct relationship (one by one relationship) between the ischemia and the present phenomenon/experiment”, one must make/add a model of “re-perfusion/stop-ischemia model” to mirror/represent the present data (partial recovery according to the ischemia degree/resolution). In my opinion, you too much emphasized the ischemia. You need not forcefully “make” an interesting story. How about simply writing the “fact”? Please reconsider. I, of course, well understand that placental ischemia is a central of the PE-pathophysiology.

Author Response

Reviewer 3 Comments:

The theme is important. Clinical usefulness is yet to be determined, this can be a candidate of therapeutic strategy of PE as the authors suggested. I have some advices.

  1. You stated, “Importantly, these systems occur in response to placental ischemia in the reduced uterine perfusion pressure (RUPP) rat model of preeclampsia. [5 6]. [2 6 7].”. In Introduction, you previously stated/touched “PE general” (mainly in human) and then here, you abruptly change the topic to “rat”. The context is inconsistent. For example, the following may be better (if your intention is so): “In PE, reduced uterine perfusion may cause placental ischemia, of which phenomenon has been well demonstrated in rat model (RUPP rat)”. The same is true to Abstract. Please make the context clear.

Response: the sentence has been revised sentence accordingly:

Importantly, in PE, reduced uterine perfusion may cause placental ischemia, of which phenomenon has been well demonstrated in the reduced uterine perfusion pressure (RUPP) rat model of preeclampsia.

Response: Abstract:  Reduced uterine perfusion in rats results in placental ischemia, which has been demonstrated in the RUPP rat model..

  1. Vaka et al. [6]; name and then citation immediately after! There are the similar mistakes here and there.

Response: We have checked and fixed citation mistakes.

    • Vaka [6] examined mitochondrial (mt) dysfunction and ROS in the RUPP rat model of PE and found that mt dysfunction contributed to the hypertension observed in response to placental ischemia
    • McCarthy [28] showed vascular mtROS and decreased respiration in HUVECs exposed to sera from PE patients compared to HUVECS cultured with sera from normal pregnant women, thus indicating the importance of the release of soluble factors in the circulation to cause cellular mt dysfunction.
  1. There have been MANY reported rat models of PE. Please state the reason why you chose this model among many others. Preferably, “to study ROS-related mechanism of PE” “the present model is theoretically better/best”; such context is better as a scientific paper. If not, “we are very accustomed to use this model and thus we used this model” (meaning) may be OK (straightforward statement is welcomed). Anyhow, please describe the reason why you used this model.

Response: we thank the reviewer for the suggestion and have indicated the reason for using the RUPP model in this study in the introduction

  1. You, all through the manuscript, emphasized “placental ischemia”. Please confirm this emphasis is theoretically correct. You used this model; actually, this model causes “placental ischemia”; however, whether this ischemia is a “single” mechanism/reason for the manifestation of PE is not demonstrated. The title also emphasizes “ischemia”. To show “a direct relationship (one by one relationship) between the ischemia and the present phenomenon/experiment”, one must make/add a model of “re-perfusion/stop-ischemia model” to mirror/represent the present data (partial recovery according to the ischemia degree/resolution). In my opinion, you too much emphasized the ischemia. You need not forcefully “make” an interesting story. How about simply writing the “fact”? Please reconsider. I, of course, well understand that placental ischemia is a central of the PE-pathophysiology.

Response: we have shown that the blood flow to the placenta following the RUPP procedure is decreased 40% which leads to ischemia. However, we do appreciate the reviewers point and have decreased the emphasis on ischemia throughout the manuscript,

Round 2

Reviewer 1 Report

I disagree with the response to my first concern. As this manuscript focused on preeclampsia, even the title also showed "preeclampsia".  I really want to see the systolic blood pressure, which is more important to MAP. The clinical definition of preeclampsia is systolic or diastolic blood pressure. In addition, only 2 rats' MAP was over 125 mmHg.  it was also no changes in blood pressure in RUPP rats.  So I believe that the model of preeclampsia was not well established, which is the big problem for this study. 

The authors did not well respond to my fourth question. If the blood pressure was only measured once, that is not enough. 

In addition, the authors did not respond to my last concern, not at all.

Reviewer 2 Report

Many thanks to the authors for providing responses and performing further experiments. Although most of the concerns have been addressed I have a few further comments which have not. 

Abstract/Introduction

PE is not defined as only new onset of hypertension - this is gestational hypertension. Please add appropriate definitation of PE or at least add organ damage to new onset hypertension.

Methods

What method/equipment was used to measure BP?

For HUVEC experiments, were sera pooled from each group or was experiments repeated 5/6 times with individual serum from each rats and data then pooled together? This needs to be clearly stated.

Results

Table 1 - there does not seem to be a statistically significant difference in %survival between NP and RUPP+LD IL-2 yet symbol "+" was added next to 99.5+/-0.1? 

State clearly that although previous study showed adverse fetal effects of LD IL-2, this study did not, which could be due to different operator's skills.

Figures 2-5 and 7 have different n numbers of rats in each group than Figure 1 especially RUPP+LD IL-2 has been reduced from n=9 to n=5-6. Why were some rats excluded from data analysis?

Figure 7 - Was there statistically significant difference between NP and RUPP LD IL-2? It seems to be...

All figure legends state t-test was used when there are always more than 2 groups being compared and therefore t test is not suitable?

Discussion

Still need to discuss what do high levels of IL-2 in your model mean- particularly in terms of the information provided in the second to last paragraph of the discussion, which states that high IL-2 levels can cause adverse pregnancy outcomes. The systemic IL-2 levels in this study also appear higher in RUPP+IL2 compared to NP so this needs to be discussed.

Author Response

Reviewer 2:

Abstract/Introduction

  1. PE is not defined as only new onset of hypertension - this is gestational hypertension. Please add appropriate definitation of PE or at least add organ damage to new onset hypertension.

Response: Thanks for your comment. We added “organ damage” to new onset hypertension as suggested to correctly reflect the proper definition of PE. It is now stated as: “PE is defined as new onset hypertension and end-organ dysfunction during pregnancy occurring after the 20th week of gestation and is associated with chronic immune activation, proteinuria, fetal growth restriction and maternal endothelial dysfunction.”

Methods

  1. What method/equipment was used to measure BP?

Response: On GD 19, blood pressure was measured with a pressure transducer (Cobe II tranducer CDX Sema) and recorded continuously for one-hour after a 30 minute stabilization period as previously described [15]. This is now more clearly stated in the methods.

  1. For HUVEC experiments, were sera pooled from each group or was experiments repeated 5/6 times with individual serum from each rats and data then pooled together? This needs to be clearly stated.

Response: Thanks for your comment, we have modified the statement to state “Cells were serum starved for 4h prior to incubation with HUVEC complete growth media and 10% of individual serum from NP (n=5), RUPP (n=4), or RUPP + LD IL-2 (n=7) sera overnight. Each experiment for individual rats were performed in duplicate and averaged together/animal. The data was then averaged for each group.”

Results

  1. Table 1 - there does not seem to be a statistically significant difference in %survival between NP and RUPP+LD IL-2 yet symbol "+" was added next to 99.5+/-0.1? 

Response: Thanks for your comment. You are correct and symbol was removed. It should have been removed on 8/10 as it says in the revision 1 edits, but it appears that it was not. Thank you for the comment.

  1. State clearly that although previous study showed adverse fetal effects of LD IL-2, this study did not, which could be due to different operator's skills.

Response: Thanks for your comment. We have revised the sentence as suggested and it now reads:” Although the previous study by Cunningham et al [14] showed adverse fetal effects of LD IL-2, this study did not, which could be due to different operator’s skills.”

  1. Figures 2-5 and 7 have different n numbers of rats in each group than Figure 1 especially RUPP+LD IL-2 has been reduced from n=9 to n=5-6. Why were some rats excluded from data analysis?

Response: Thanks for your comment. The rats were not excluded from data analysis there were actually processed and intended for use in the analysis, however, in performing mt function experiments, sometimes there is degradation of tissue, mt isolation, or additional problems during the isolation of mitochondria from the renal or placental tissues that cause disruption of the mitochondrial membrane and therefore, entact mitochondria were collected and this rat then has to be excluded from the data analysis.

  1. Figure 7 - Was there statistically significant difference between NP and RUPP LD IL-2? It seems to be...

Response: There was no statistically significant difference between NP and RUPP+ LD IL-2 based on the results from the one-way ANOVA.

  1. All figure legends state t-test was used when there are always more than 2 groups being compared and therefore t test is not suitable?

Response: Thank you for your comment. The figure legends were not correctly stated, and they have been reviewed/edited to reflect the correct statistical analyses that were performed. Results were reported as means ± SEM and considered statistically significant when p<0.05.

Discussion

  1. Still need to discuss what do high levels of IL-2 in your model mean- particularly in terms of the information provided in the second to last paragraph of the discussion, which states that high IL-2 levels can cause adverse pregnancy outcomes. The systemic IL-2 levels in this study also appear higher in RUPP+IL2 compared to NP so this needs to be discussed.

Response: Thank you for your comment. We have discussed what the high levels of IL-2 mean in response to our model in the second to last paragraph of the discussion.

Round 3

Reviewer 1 Report

The current definition of preeclampsia recommended by ACOG and ISSHP is systolic blood pressure greater than 140mmHg or diastolic blood greater than 90mmHg, with other laboratory data. This definition is well-accepted and well used in all related studies. As I mentioned before, the blood pressure in the rat model presented in this manuscript was not acceptable.MAP at 125 mmHg is a cut off point for preeclampsia in animal model. This indicated that the model was not well established by the authors. Therefore I cannot support this manuscript for being published in Cells. 

Author Response

Reviewer 1: The current definition of preeclampsia recommended by ACOG and ISSHP is systolic blood pressure greater than 140mmHg or diastolic blood greater than 90mmHg, with other laboratory data. This definition is well-accepted and well used in all related studies. As I mentioned before, the blood pressure in the rat model presented in this manuscript was not acceptable. MAP at 125 mmHg is a cut off point for preeclampsia in animal model. This indicated that the model was not well established by the authors. Therefore I cannot support this manuscript for being published in Cells. 

Dear Reviewer 1 please see screenshot of pubmed search of the RUPP model below. With all due respect there have been over 800 publications using this model in the rat as a model of preeclampsia. These studies were performed by ours and other laboratories and this is a well accepted model of preeclampsia even though the systolic and diastolic numbers may not always be 140/90. In fact there are women whose blood pressures are not 140/90 but are diagnosed with PE.  Moreover the purpose of this study was not to establish the RUPP as a model of PE but to learn more about the effect IL-2 and its role in the context of inflammation in response to placental ischemia.
